# Integrative Map of *HIF1A* Regulatory Elements and Variations

**DOI:** 10.3390/genes12101526

**Published:** 2021-09-28

**Authors:** Tanja Kunej

**Affiliations:** Department of Animal Science, Biotechnical Faculty, University of Ljubljana, 1000 Ljubljana, Slovenia; tanja.kunej@bf.uni-lj.si; Tel.: +386-1-2303-890

**Keywords:** cancer, erythrocytosis, hypoxia-inducible factor (HIF), *HIF1A*, hypoxia

## Abstract

Hypoxia-inducible factor (HIF) family of transcription factors (HIF1A, EPAS1, and HIF3A) are regulators of the cellular response to hypoxia. They have been shown to be involved in development of various diseases such as cancer, diabetes, and erythrocytosis. A complete map of connections between HIF family of genes with various omics types has not yet been developed. The main aim of the present analysis was to construct the integrative map of genomic elements associated with *HIF1A* gene and prioritize potentially deleterious variants. Various genomic databases and bioinformatics tools were used, including Ensembl, MirTarBase, STRING, Cytoscape, MethPrimer, CADD, SIFT, and UALCAN. Integrative *HIF1A* gene map was visualized and includes transcriptional and post-transcriptional regulators, downstream targets, and genetic variants. One CpG island overlaps transcription start site of the *HIF1A* gene. Out of over 450 missense variants, four have predicted deleterious effect on protein function by at least five bioinformatics tools. Currently there are 85 miRNAs reported to target *HIF1A*. HIF1A downstream targets include protein-coding genes, long noncoding RNAs, and microRNAs (hypoxamiRs). The study presents the first integration of heterogeneous molecular interactions associated with *HIF1A* gene enabling a holistic view of the gene and lays the groundwork for supplementing the data in the future.

## 1. Introduction

Hypoxia-inducible factor (HIF) family of transcription factors are master regulators of the cellular response to low oxygen availability (hypoxia). They coordinate hypoxic response in cells, ensuring metabolic and vascular adaptation to shortage of oxygen. Heterodimeric HIF transcription factors consist of the α and β subunits. The α subunit of the HIF heterodimer is O2-sensitive, and in hypoxia, it functions as a master regulator of various genes involved in hypoxia pathway. The β subunit of HIF heterodimer (HIF1B, HIF-1β) with the official gene name aryl hydrocarbon nuclear translocator (ARNT) is constitutively expressed [1,2].

HIF α paralogs dimerize with the same β subunit (HIF-1β). Three paralogs of α subunit are known: hypoxia-inducible factor 1 subunit α (HIF1A, HIF-1α), endothelial PAS domain-containing protein 1 (EPAS1, HIF2A, HIF-2α), and hypoxia-inducible factor 3 subunit α (HIF3A; Figure 1). Among HIF family of genes HIF1A and EPAS1 genes have been a topic of several studies, however, HIF3A is less studied [3,4].

Under normoxia, the HIF signaling pathway is inhibited due to degradation of HIFα subunits. Two proline residues of HIF1A and EPAS1 are hydroxylated by prolyl hydroxylase domain protein 2 (PHD2), also named egl-9 family hypoxia-inducible factor 1 (EGLN1). Modification by prolyl hydroxylation is a regulatory event that targets HIF subunits for ubiquitin-mediated proteolysis via binding to the Hippel–Lindau tumor suppressor (VHL). In hypoxic conditions, the HIF α subunit is activated and stabilized, resulting in activation of HIF target genes and various activities (Figure 2) [5].

The bHLH-PAS motifs at N-terminus are essential for heterodimer formation between HIF α and β subunits and for binding to the sequence on the target genes. Conserved HLH domain is involved in DNA binding to hypoxia response element (HRE) in various target genes. HIF α paralogs have both unique and common downstream targets in hypoxic gene regulation. The HIF1A and EPAS subunits have two transactivation domains (TAD), N-TAD and C-TAD, which are responsible for transcriptional activity. C-TAD interacts with co-activators such as CBP/p300 to modulate gene transcription under hypoxia. N-TAD is responsible for stabilizing HIFα against degradation. HIFα subunits have an oxygen-dependent degradation (ODD) domain (ODDD) overlapping N-TAD. The ODDD is important in mediating O_2_ regulation stability [6,7].

Asparagine 803 located in the transactivation domain is hydroxylated in oxygenated cells by factor inhibiting HIF-1 (FIH-1; HIF1AN; hypoxia-inducible factor 1 subunit α inhibitor), which blocks the binding of the co-activators p300 and CBP (EP300; E1A binding protein/CREBBP; CREB binding protein) [7]. Under hypoxic conditions, the prolyl and asparaginyl hydroxylation reactions are inhibited by oxygen deprivation [8,9].

HIF genes have been shown to be involved in the development of various diseases such as cancer, diabetes, and erythrocytosis [7,10]. Inhibiting the interaction between transcription factor HIF-1α and coactivator p300/CBP represents one of the possible approaches for blocking hypoxia pathway in tumors [11].

HIF genes are involved in the complex interplay of molecular interactions at various omics levels, including genomics, transcriptomics, proteomics, epigenomics, and miRNomics. Gene regulatory networks (GRN) are sets of proteins and RNAs that interact to control the level of expression of various genes. The main players in regulatory networks are DNA-binding protein transcription factors as they modulate the first step in gene expression. Upstream regulators include transcription factors (TFs) and any gene or small molecule that has been observed experimentally to affect gene expression (Li et al., 2015).

Various epigenetic mechanisms regulate gene expression, including DNA methylation/hydroxymethylation, post-translational modifications of histones, chromatin remodeling, and regulation by noncoding RNAs. Noncoding RNAs are divided into two major groups, short ncRNA (for example miRNAs) and long noncoding RNAs (lncRNAs) [12]. MicroRNAs that are under the control of HIF genes are termed hypoxia-regulated miRNAs (HRMs) or hypoxia-induced miRNAs (hypoxamiR)—for example, miR-210. Hypoxia is therefore a powerful stimulus regulating the expression of a specific subset of miRNAs, which are regulators of the cell responses to decreased oxygen tension. miR-210, for example, is elevated in patients with ischemic diseases and most solid tumors [13].

The number of publications on HIF transcription factor family is increasing, and therefore efforts toward integration and organization of published knowledge are an urgent need to facilitate research developments of this study field. A map of HIF regulatory elements would enable more planned and coordinated development of this topic. The aim of the present study was therefore to review the main elements involved in the molecular interplay associated with *HIF1A* gene and visualize the information as the map of regulatory elements including upstream regulatory elements regulating *HIF1A* expression, post-transcriptional regulators, and its downstream targets.

## 2. Materials and Methods

Relevant genomics data regarding the *HIF1A* gene in humans was extracted from genomics databases and publications. The terminology for the gene names and symbols was edited according to the HUGO gene nomenclature committee (HGNC) (https://www.genenames.org/ (accessed on 2 August 2021). The *HIF1A* gene structure and genetic variations were obtained from the Ensembl genome browser [14], release 104 (May 2021). Prediction of the effect of genetic variants on protein function was performed using six bioinformatics tools integrated in the Ensembl database: SIFT (sorting intolerant from tolerant), PolyPhen (polymorphism phenotyping), CADD (combined annotation-dependent depletion), REVEL (rare exome variant ensemble learner), MetaLR, and Mutation Assessor. Experimentally validated microRNA-target (MTI) interactions were obtained from the MirTarBase [15] and visualized using Cytoscape software [16]. Protein–protein interaction (PPI) networks were obtained from the STRING tool [17] using the setting network type/physical network, in which the edges indicate that the proteins are part of a physical complex. The MethPrimer tool was used for CpG island prediction using the following criteria: CpG island size >200, GC percent >50.0, and observed/expected CpG ratio >0.6 [18]. Phenotype associations were extracted from the Ensembl database, which imports data from the Cancer Gene Census (CGC; https://cancer.sanger.ac.uk/census) (accessed on 2 August 2021)) [19], Zebrafish Information Network (ZFIN; https://zfin.atlassian.net/wiki/spaces/general/overview?mode=global (accessed on 2 August 2021)) [20], Rat Genome Database (RGD; https://rgd.mcw.edu/ (accessed on 2 August 2021)) [21], and Mouse Genome Informatics (MGI; http://www.informatics.jax.org/ (accessed on 2 August 2021) [22]. Cancer OMICS data were extracted from the UALCAN interactive web resource, which includes the Cancer Genome Atlas (TCGA) and the Clinical Proteomic Tumor Analysis Consortium (CPTAC) data [23,24]. The UALCAN database also includes the TargetScan tool (release 7.2), which was used for the prediction of the miRNA target sites within the 3′ UTR region of the *HIF1A* gene [25].

## 3. Results and Discussion

### 3.1. Characteristics of the HIF1A Gene

Human *HIF1A* gene is associated with the following gene identifiers (Gene IDs): ENSG00000100644 (Ensembl ID), 3091 (NCBI ID), and 4910 (HGNC ID). According to the latest release of the Ensembl database, the *HIF1A* gene has 12 annotated transcripts (splice variants), with the following biotypes: four protein-coding, two with retained intron, and six processed transcripts—transcripts that do not contain an open reading frame (ORF). Two protein-coding transcripts are marked as the golden path (merged Ensembl/Havana): ENST00000323441.10 (HIF1A-201) and ENST00000337138.9 (HIF1A-202). The longest transcript, ENST00000337138.9, consists of 15 coding exons, the transcript length is 3946 bps, the translation length is 826 residues, and it is associated with 14,136 variant alleles (Figure 3). The shorter transcript ENST00000323441.10 has 14 exons, the transcript length is 3555 bp, the translation length is 826 aa, and it is associated with 14,018 variant alleles. In humans, the *HIF1A* gene has seven paralogs: *NPAS4*, *EPAS1*, *NPAS3*, *NPAS1*, *SIM1*, *SIM2*, and *HIF3A*. The *HIF1A* gene currently has 226 annotated orthologues—for example, *Hif1a* (ENSMUSG00000021109) in mouse, *hif1aa* (ENSDARG00000006181) in zebrafish, and *HIF1A* (ENSBTAG00000020935) in cow.

### 3.2. Genetic Variability of the HIF1A Gene

The *HIF1A* gene includes genetic variants with various biotypes, including missense and synonymous variants present in all exons, and frameshift variants in exons 2, 5, 8, 10, 11, and 14 (Figure 3). Several splice variants, marked in orange in Figure 3, are present in the *HIF1A* gene. Splice variants overlap splice sites at the exon/intron borders and are defined as the changes located either within 1–3 nucleotides of the exon or 3–8 nucleotides of the intron. Besides several intronic splice variants, there are also splice variants located within exons; within the first three nucleotides of exons 5, 11, 13, and 14 and within the last three nucleotides of exon 2. Additionally, the *HIF1A* gene also has three splice donor variants, located within the first two nucleotides of the intron (introns 6, 7, and 11) and one splice acceptor variant located in the last two nucleotides of the introns (introns 7 and 12). The *HIF1A* gene currently does not have variants within the start codon. However, two variants are located in the stop-codon: stop lost (rs1389320297; */R) and stop retained variant (rs770333662; */*). The gene also has nine stop gained variants in exons 8, 9, 10, and 15.

Figure 4 presents the *HIF1A* nucleotide sequence of exon 12 with marked genetic variants. The location of two frequently studied missense variants at the protein positions 582 and 588 are marked: rs11549465 (p.Pro582Ser) and rs11549467 (p.Ala588Thr). These two variants are located within the ODD domain.

The analysis for the presence of variants located at the sites of protein hydroxylation revealed that there are variations located at all three sites: P402, P564, and N803. Two synonymous variations encoding for same amino acid are present at the site of proline hydroxylation by HIF prolyl hydroxylase EGLN1 (P402 and P564) and a missense variant is located at the site of asparagine hydroxylation (N803). The first site of proline hydroxylation (P402) is located in exon 9, and there is a synonymous variant within this codon (rs750187829 A > G; codons CCA > CCG; Figure 3). The second site of proline hydroxylation is located in exon 12; at the proline residue at position 564 (P564). A genetic variant, rs41492849 C > T, is located within this codon (Figure 4). Additionally, the missense variation rs1220369809 T > G is located at the asparagine 803 (N803) in exon 15, the site of hydroxylation by HIF1AN. This variation is predicted to have a deleterious effect on protein function or, it is likely disease-causing by four bioinformatics tools: SIFT, PolyPhen, REVEL, and MetaLR.

The current release of the Ensembl database lists over 450 missense variants of the *HIF1A* gene with calculated predictions on protein function. The missense variant rs1566567018 G > A at protein position 29 has been predicted to have a deleterious effect by all six tools: deleterious (SIFT), probably damaging (PolyPhen), likely deleterious (CADD), likely disease-causing (REVEL), damaging (MetaLR), and high functional impact (Mutation Assessor). Additionally, three polymorphisms that have predicted a deleterious effect using five bioinformatics tools are located on protein position 27 (rs754062510), 29 (rs746540920), and 129 (rs1594871023). Examples of five *HIF1A* variants with a high predicted deleterious effect using bioinformatics tools are shown in Figure 5. These five polymorphisms are located at protein position 17 (rs904387412 and rs1163104034), position 29 (rs1566567018), position 129 (rs1594871023), and position 545 (rs1339195532).

### 3.3. A Map of HIF Regulatory Elements

The *HIF1A* gene has been studied at various omics levels, including genomics (DNA level), transcriptomics (RNA level), proteomics, epigenomics, interactomics, and miRNomics levels. The *HIF1A* gene is involved in a complex interplay of interactions, and a subset of regulatory elements associated with the *HIF1A* gene is shown in Figure 6. HIF1A acts as a transcription factor of several downstream targets; however, HIF1A is under the control of various transcription factors (TFs) binding to transcription factor binding sites (TFBS). Additionally, miRNA regulate the expression of the HIF1A gene through interaction of their seed region with the target region, primarily at the 3′ UTR region. The HIF1A transcription factor drives the expression of a large set of target genes, including protein-coding genes, long noncoding RNAs (lncRNAs), and short ncRNAs, including miRNAs (HRMs, hypoxamiRs).

#### 3.3.1. HIF Target Genes

The *HIF* family of genes has been shown to target protein-coding genes as well as several classes of noncoding RNAs. The number of reported HIF target genes exceeds 1000 and is increasing [26,27]; however, a complete database of target genes does not yet exist. The preliminary catalog of HIF1A targets consists of 98 genes that were described in 51 published papers. Those target genes were shown to be associated with 20 pathways, including metabolism of carbohydrates and pathways in cancer. Additionally, reanalysis of genomic coordinates of hypoxia response elements (HREs) revealed six polymorphisms within HRE sites (termed as HRE-SNPs) of four genes: *ABCG2*, *ACE*, *CA9*, and *CP* [28]. The list of target genes is not yet complete, and further studies are needed to reveal the complete HIF1A targetome. Large heterogeneity of result presentation in scientific literature reporting HIF1A targets was observed, and therefore a minimal checklist for reporting HIF1A targets was suggested; it consisted of 10 relevant data types: official symbols of the target genes, species, methodology, cell lines, expression of the target gene, genomics location of HRE sites, presence of HRE polymorphisms, associated phenotypes or diseases, and available identification numbers of genes, diseases, and species [28]. Standardized reporting of HIF targets would facilitate data curation and enable faster development of a complete catalog of HIF targetome.

The hypoxic regulatory network includes transcripts of protein-coding genes and a class of long noncoding RNAs termed T-UCRs (transcribed ultraconserved regions). The hypoxia-induced noncoding ultraconserved transcript (*HINCUT1*) is located in the retained intron of the *OGT* gene and has been shown to be upregulated in colon cancer and involved in protein glycosylation in hypoxia [29]. HIF1A also promotes the expression of several hypoxamiRs, including miR-210, miR-146a, miR145, miR-382, miR-191, miR-363, miR-421 in tumor cells, miR-204 in neuronal cells, and miR-30a and miR-21 in cardiomyocytes (reviewed in [30]).

#### 3.3.2. Upstream Regulation of HIF1A Signaling

Various pathways have been shown to regulate HIF activity by regulating HIF synthesis [6]. The HIF family of genes is affected by various factors, including hormones, growth factors, and cytokines on transcriptional and post-transcriptional levels [3]. For example, the PI3K/Akt/mTOR pathway plays a significant role in regulating HIF activity. Phosphoinositide 3-kinase (PI3K), pyruvate dehydrogenase kinase (PDK), and protein kinase B (PKB; Akt) activation induced by growth factors (GFs) activate the mammalian target of the rapamycin (mechanistic target of rapamycin kinase; mTOR) pathway, which results in elevated HIF-1α transcriptional activity [6]. Other pathways/factors controlling HIF regulation include the TNF signaling/NF-κB (nuclear factor kappa-B) pathway, RAS/RAF/MEK/ERK kinase cascade, mouse double minute 2 homolog (MDM2), and heat shock protein 90 (HSP90) [3,6].

#### 3.3.3. Epigenomics

The *HIF1A* gene is under the control of various epigenetic mechanisms, including DNA methylation, histone modifications, and miRNAs. A large CpG island comprising 1431 bp is present around the transcription start site of the *HIF1A* gene (Figure 7). Hypomethylation non-CpG/CpG sites in the promoter of the *HIF1A* gene in combination with increased H3K9Ac modification was shown to contribute to increased transcription and expression in breast cancer [31].

Ten-eleven translocation (TET) methylcytosine dioxygenases catalyze the conversion of 5-mC to 5′-hmC. TET1-mediated 5-hmC changes have been shown to be an epigenetic component of the hypoxic response. Hypoxia increases global 5-hmC levels, and accumulation of 5-hmC sites has been identified at hypoxia response genes. 5-hmC gains are also located at HRE sites, facilitating DNA demethylation and HIF binding. Hypoxia results in transcriptional activation of TET1, and induction of hypoxia-responsive genes and global 5-hmC increases require TET1. 5-hmC increases and TET1 upregulation in hypoxia are HIF1 dependent [32].

Lysine methyltransferases G9a and G9a-like protein (GLP) methylate HIF1A protein and inhibit HIF1A activity within solid tumors, making it unable to bind to the HRE of its target genes, resulting in inhibition of the downstream HIF pathway. G9a and GLP catalyze mono- and di-methylation of HIF-1α at lysine (K) 674 in vitro and in vivo [3,33]. Interestingly, the present analysis revealed that the missense polymorphism is located at this position (rs779328210, Lys674Asn) and might affect regulation of HIF1A activity.

It has been reported that HIF1A employs diverse cofactors to regulate different subsets of target genes. For example, HIF1A acts on KAT5 (lysine acetyltransferase 5; TIP60), which leads to chromatin histone acetylation and then to the activation of polymerase II, which activates the transcription of HIF-1α target genes [34].

#### 3.3.4. MicroRNA Regulation

*HIF1A* has been shown to be under the control of several miRNAs. Based on the miRTarBase, a genomic resource of experimentally validated MTIs, there are currently 85 miRNAs reported to be in interaction with the *HIF1A* gene (Figure 8). Some interactions between miRNAs and *HIF1A* mRNA have been reported in more than one publication and 116 miRNA-*HIF1A* interactions have been described in 45 scientific papers. Most of these interactions were identified using methodology considered as weak validation status, such as photoactivatable ribonucleoside-enhanced crosslinking and immunoprecipitation (PAR-CLIP), crosslinking, ligation, and sequencing of hybrids (CLASH), high-throughput sequencing of RNA isolated by crosslinking immunoprecipitation (HITS-CLIP), or qRT-PCR. However, 36 of the reported interactions have a strong validation status and were confirmed using approaches such as luciferase reporter assay or western blot.

#### 3.3.5. Protein Interactions

Figure 9a presents the PPI network associated with HIF1A protein using the STRING tool. The network includes 11 nodes, and it has significantly more interactions than expected (PPI enrichment *p*-value: 5.39 × 10^−6^). The HIF1A PPI network consists of the following interacting partners: EGLN3, ARNT, ARNT2, EP300, VHL, CREBBP, TCEB1, TCEB2, CUL2, and EGLN1. Adding more nodes to the network resulted in the extended network consisting of 21 proteins (Figure 9b). An extensive list of additional HIF1A protein interacting partners was reviewed by Semenza, and the list continues to grow rapidly [2].

#### 3.3.6. Post-Translational Modifications (PTMs)

Hydroxylation has been described as the major modification of the HIF-α subunits. Additionally, several other post-translational modifications play a role in HIF localization, stability, and activity: acetylation, methylation, S-nitrosylation, phosphorylation, sumolation, and ubiquitination [2,35].

### 3.4. Association of HIF1A with Diseases

Genetic variations and dysregulation of the *HIF1A* gene have been shown to be associated with the development of several diseases. Our previous review of the literature revealed 16 SNPs with significant associations with 40 different phenotypes, including 6 SNPs associated with 14 cancer types [10]. Genetic variants of the *HIF1A* gene have been shown to be associated with cardiovascular system diseases such as ischemic heart disease, coronary artery disease (CAD), premature coronary artery disease, pre-eclampsia, and acute myocardial infarction. Moreover, *HIF1A* SNPs have been shown to be involved in metabolic disorders such as diabetic nephropathy. Missense SNPs rs11549465 (p.Pro582Ser) and rs11549467 (p.Ala588Thr) within the ODD domain are most frequently studied.

According to the Ensembl/Cancer Gene Census database, the *HIF1A* gene is associated with 62 phenotypes in humans (Appendix A). Additionally, orthologues of this gene in other species have also been associated with several phenotype, disease, and trait annotations. HIF1A–phenotype associations from the Ensembl browser, which integrates the data from ZFIN, MGI, and RGD, are shown in Appendix A.

Appendix A presents HIF1A-associated data, extracted from the UALCAN database, an interactive web resource for analyzing cancer omics data: (1) expression of *HIF1A* across TCGA cancers with tumor and normal samples (Appendix A), (2) effect of *HIF1A* expression level on bladder urothelial carcinoma (BLCA) patient survival (Appendix A), (3) HIF1A proteomic expression profile based on sample types (Appendix A), and (4) conserved miRNA mir-199-5p target site within the 3′ UTR region of the *HIF1A* gene (Appendix A). The present study presents only part of the HIF1A data available in the UALCAN database, and additional in-depth study could contribute to revealing the unknown mechanisms of HIF1A involvement in the development of various cancer types.

Imbalance in molecular mechanisms of the hypoxia-inducible factor-erythropoietin (HIF-EPO) pathway can result in hematological disorders [36]. Regulation of the *EPO* gene by HIF raises the possibility that variants in all three HIFA paralogs play a role in disorders of erythropoiesis, such as erythrocytosis. *EPAS1* was shown to be a key regulator of EPO production, and variants of the *EPAS1* gene have been reported as a cause of familial erythrocytosis type 4 (ECYT4) [7]. To date, 20 *EPAS1* variants identified in patients with erythrocytosis or associated with its symptoms are known [7]. By contrast, only two variants of the *HIF3A* gene are associated with FE [37]. The missense variant p.Pro582Ser located in the *HIF1A* gene was found to be associated with higher Hb and ferritin levels in male blood donors. This polymorphism affects red blood cell and iron homeostasis after blood loss and is associated with a resistance to anemia in males [38]. It was also shown that the Pro582Ser variant does not impair HIF1A prolyl hydroxylation in the ODD domain and it does not diminish the association of HIF1A with VHL [39]. Current literature shows that *EPAS1* has an important role in erythropoiesis; however, the role of *HIF1A* and *HIF3A* in the development of erythrocytosis should be further investigated [7].

### 3.5. Future Developments

The present data synthesis on the *HIF1A* gene presents the first step toward the construction of a map of regulatory elements for the *HIF1A* gene and aims to establish a holistic view of the *HIF1A* gene integrating data from various genomics resources and publications. However, the integrative map is not yet complete, and additional data should be integrated from published and upcoming studies.

Some of research directions may include:Update of the review on associations between *HIF* genes and diseases.Development of a catalog of regulatory elements regulating *HIF1A* expression (upstream regulators).Update of the catalog of reported HIF1A target genes (downstream signaling cascades).Identification of novel HIF1A target genes.Multi-omics view in understanding HIF1A regulation, including genomics–DNA level, transcriptomics–RNA level, proteomics, glycomics, epigenomics, and miRNomics.Identification of therapeutic strategies targeting the HIF signaling pathway for therapy in various diseases, including cancer.Analysis of potential functional effect of polymorphisms located within sites of HIF1A protein hydroxylation and methylation.

## 4. Conclusions

The main aim of this study was to summarize the current state of the *HIF1A* gene knowledge and discuss future directions. The current data synthesis is the first step toward creating a map of regulatory factors for the *HIF1A* gene, with the intention of building a holistic perspective of the *HIF1A* gene by combining data from publications and diverse genomics resources such as Ensembl, STRING, and UALCAN. The summary of genomic databases and bioinformatics tools used in the present analysis is presented in Appendix A. Out of over 450 missense variants, four have predicted a deleterious effect on protein function by at least five bioinformatics tools. Additionally, 85 miRNAs have been reported to target *HIF1A*. HIF1A downstream targets include protein-coding genes, long noncoding RNAs, and miRNAs. Because of the increasing research interest in the HIF family of genes, systematic reviews are needed in the future. Development of databases including all known molecular interactions and genetic variations associated with HIF1A would substantially contribute to the development of the field.

One of the major obstacles that hinder the development of HIF1A research is the heterogeneity of published results. The knowledge related to the *HIF1A* gene is presently scattered across various publications and databases and does not enable a systems view of this research topic. For a better understanding of the research field, an integration of these diverse data is needed and a complete map regulatory elements associated to all HIF paralogs should be developed.

## Figures and Tables

**Figure 1 genes-12-01526-f001:**
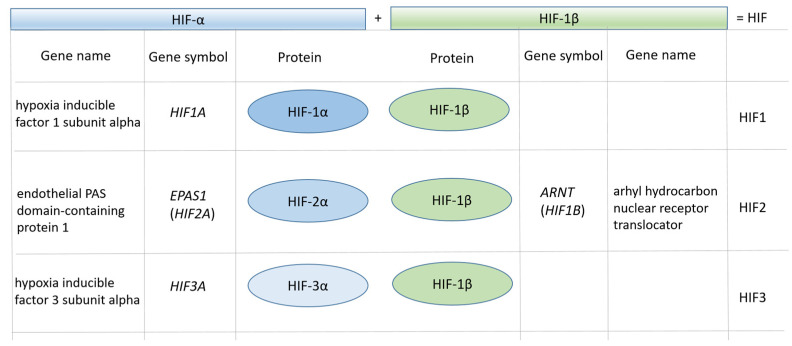
Terminology of the HIF gene family.

**Figure 2 genes-12-01526-f002:**
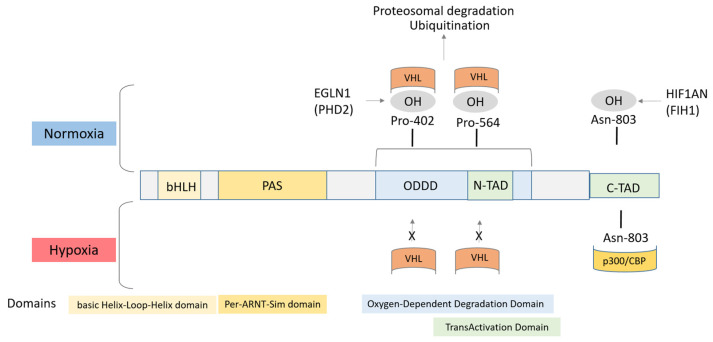
Structure of the HIF1α protein. bHLH: basic helix–loop–helix; PAS: Per-ARNT-Sim domain; ODDD: oxygen-dependent degradation domain; TAD: transactivation domain.

**Figure 3 genes-12-01526-f003:**
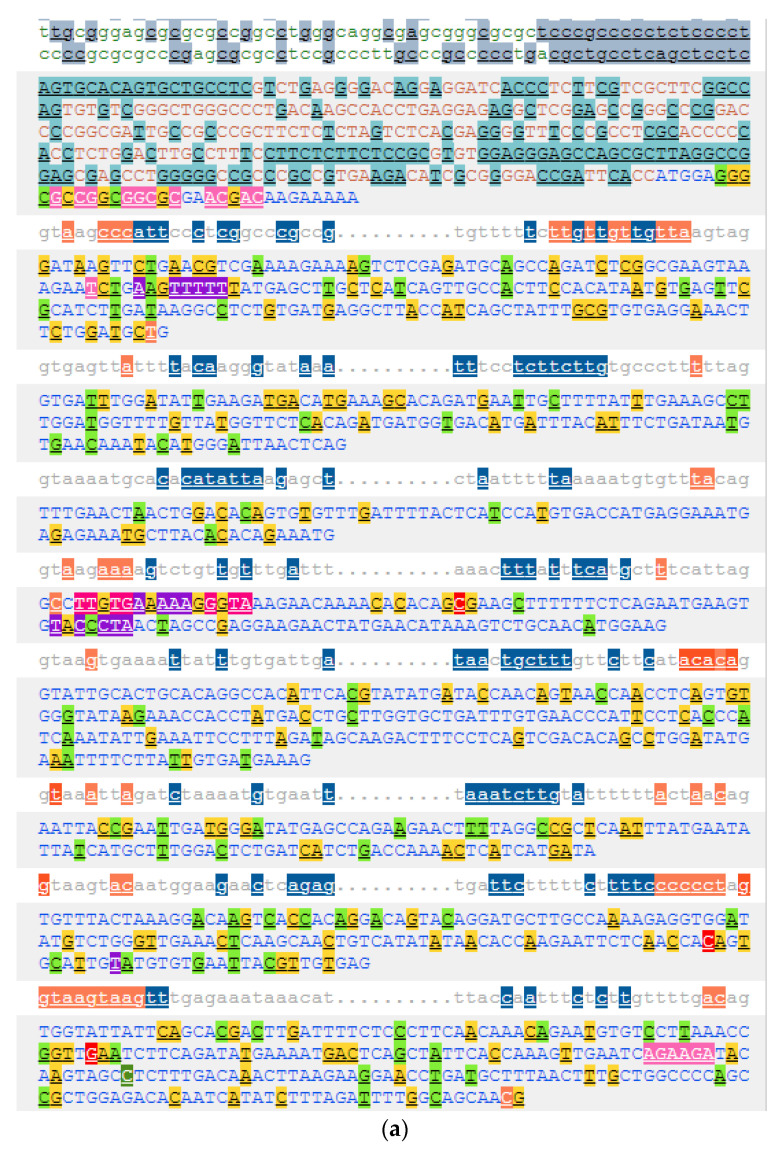
Human *HIF1A* gene structure and genetic variants. (**a**) Gene sequence from 5′ UTR/exon 1 to exon 9. (**b**) *HIF1A* gene sequence from intron 9 to exon 15/3′ UTR. Partial sequences of introns are shown; start and end. (**c**) Color legend. The figures were obtained from the Ensembl genome browser.

**Figure 4 genes-12-01526-f004:**
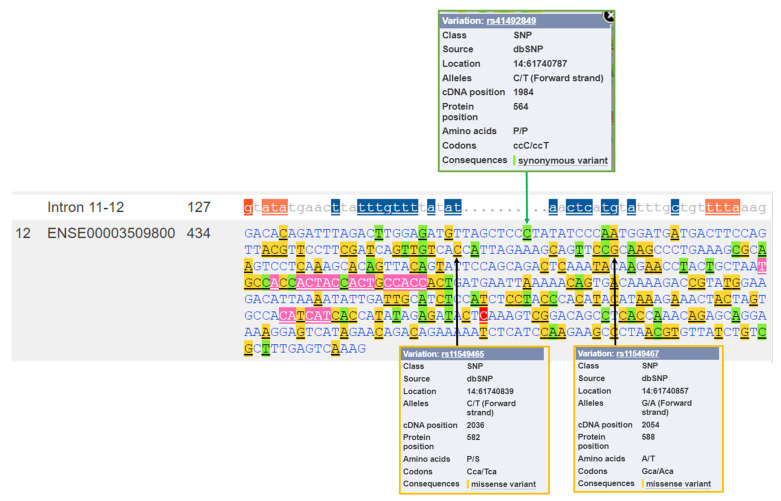
*HIF1A* nucleotide sequence of exon 12 with marked genetic variants. The green arrow denotes a synonymous variant C > T at the proline residue at position 564 (P564); the site of hydroxylation by HIF prolyl hydroxylase EGLN1. The black arrows denote missense variants at the protein positions 582 and 588. Legend: P/S: Pro/Ser; A/T: Ala/Thr.

**Figure 5 genes-12-01526-f005:**
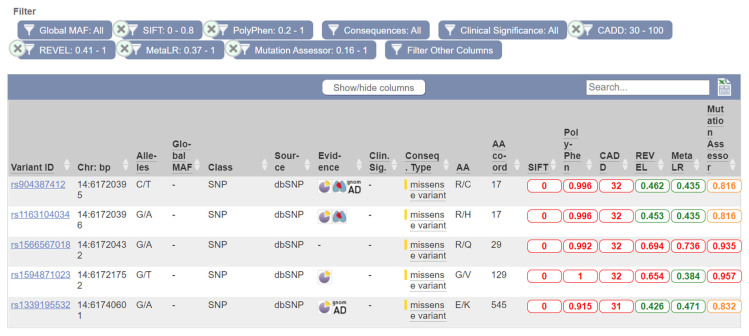
Five *HIF1A* polymorphisms with predicted effect on protein function. The analysis was performed using six bioinformatics tools included in the Ensembl genome browser.

**Figure 6 genes-12-01526-f006:**
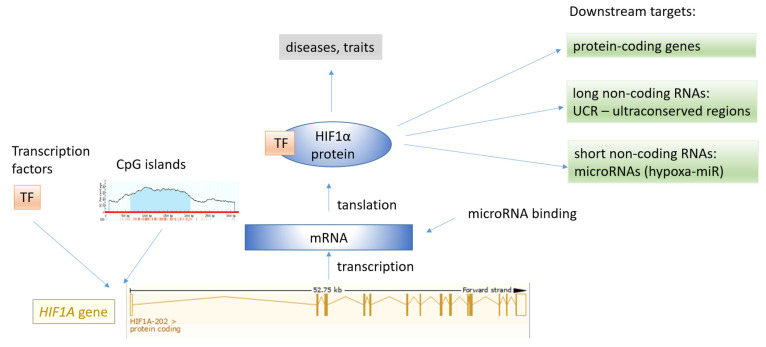
The map of regulatory elements associated with the HIF transcription factor family. The *HIF1A* gene structure was obtained from the Ensembl genome browser.

**Figure 7 genes-12-01526-f007:**
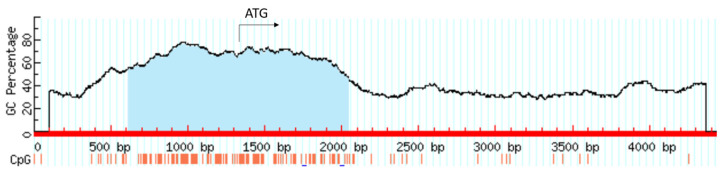
CpG island comprising 1431 bp associated with the *HIF1A* gene.

**Figure 8 genes-12-01526-f008:**
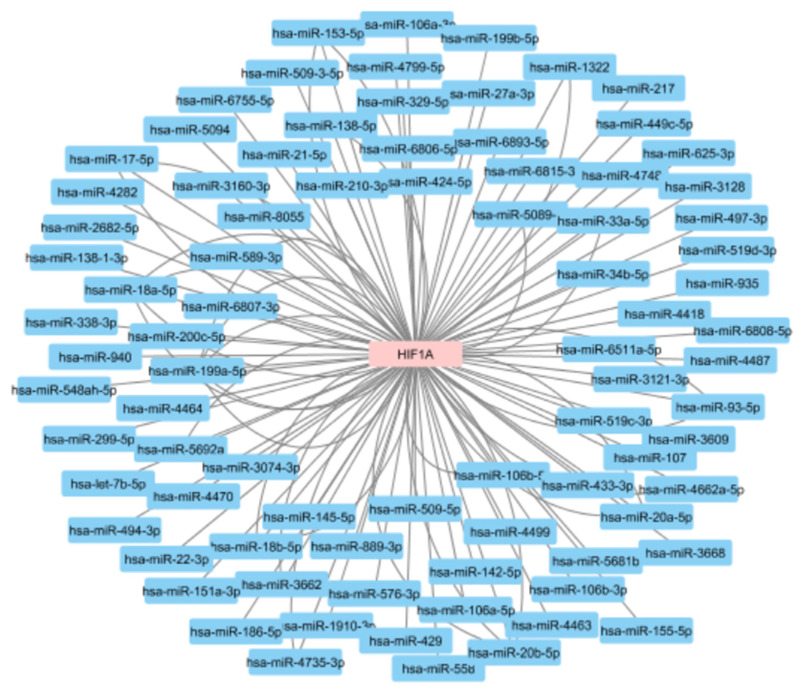
MicroRNAs reported to regulate the *HIF1A* gene. The data obtained from the miRTarBase.

**Figure 9 genes-12-01526-f009:**
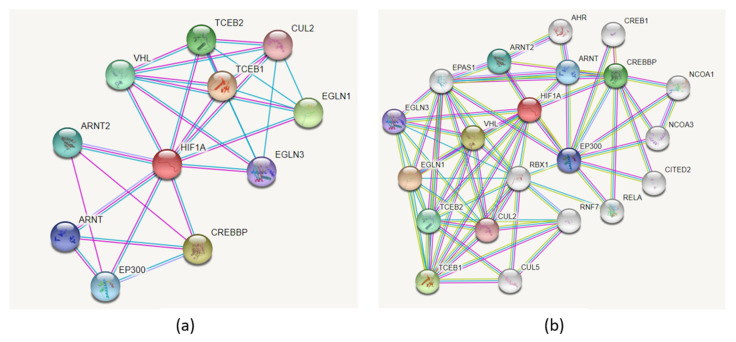
The PPI network associated with HIF1A protein visualized using the STRING tool. (**a**) HIF1A direct interacting partners; the network includes 11 nodes. (**b**) HIF1A extended interaction network with first neighbors consisting of 21 proteins.

## Data Availability

For preparation of the manuscript, the following publicly available databases were used: HGNC (https://www.genenames.org/) (accessed on 2 August 2021). Ensembl genome browser (https://www.ensembl.org/index.html) (accessed on 2 August 2021). MirTarBase (https://mirtarbase.cuhk.edu.cn/) (accessed on 2 August 2021) STRING (https://string-db.org/) accessed on 2 August 2021. MethPrimer (https://www.urogene.org/methprimer/) accessed on 2 August 2021. UALCAN (http://ualcan.path.uab.edu/index.html) accessed on 20 September 2021. All the data are presented within the article and in the Appendix A.

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
