# Peer review of "Integrative Map of HIF1A Regulatory Elements and Variations"

_genes, 2021, doi:10.3390/genes12101526_

Round 1

Reviewer 1 Report

The manuscript presented for review describes the  study aimed was  to  construct  the  integrative  map  of  genomic  elements  associated with  HIF1A  gene  and  prioritize  potentially  deleterious  variants.
 The author of the manuscript for data analysis used various genomic databases and bioinformatics tools, including Ensembl, MirTarBase, STRING, Cytoscape, MethPrimer, CADD, SIFT, and PolyPhen.

I propose to extend the section "3.4. Association of HIF1A with diseases" with information related of HIF1A gene and protein expression in cancer. The analysis should be carried out on the basis of generally available bioinformatics databases containing data from The Cancer Genome Atlas (TCGA) and the Clinical Proteomic Tumor Analysis Consortium (CPTAC).It would also be good to do a survival analysis (Kaplan-Meier curves) - select the most important information from the analysis.

The author writes that miRNA-HIF1A interactions have been described in many publications. I also propose to try to analyze the information contained in the TCGA miRNA database for the most important MicroRNAs regulating the HIF1A gene (for example in the UALCAN or other database) - select the most important information from the analysis.

The conclusions are insufficient and too general. The conclusions should contain specific information. The author should summarize the results of his analysis and should indicate what the results of this analysis show (as evidenced by the research results, what the obtained knowledge contributes to the scientific field). 

Figure 3 is out of focus - please correct it.

Author Response

Reviewer 1

The manuscript presented for review describes the  study aimed was  to  construct  the  integrative  map  of  genomic  elements  associated with  HIF1A  gene  and  prioritize  potentially  deleterious  variants.  The author of the manuscript for data analysis used various genomic databases and bioinformatics tools, including Ensembl, MirTarBase, STRING, Cytoscape, MethPrimer, CADD, SIFT, and PolyPhen.

I propose to extend the section "3.4. Association of HIF1A with diseases" with information related of HIF1A gene and protein expression in cancer. The analysis should be carried out on the basis of generally available bioinformatics databases containing data from The Cancer Genome Atlas (TCGA) and the Clinical Proteomic Tumor Analysis Consortium (CPTAC).It would also be good to do a survival analysis (Kaplan-Meier curves) - select the most important information from the analysis. The author writes that miRNA-HIF1A interactions have been described in many publications. I also propose to try to analyze the information contained in the TCGA miRNA database for the most important MicroRNAs regulating the HIF1A gene (for example in the UALCAN or other database) - select the most important information from the analysis.

Answer:

I would like to thank very much the reviewer for suggesting to extract relevant cancer data from the UALCAN database which among other incorporates TCGA, CPTAC data and TargetScan tool. The extracted information were presented in the Figures S1A-D. Two references citing the UALCAN database were also included in the manuscript:

  1. Chen, F.; Chandrashekar, D.S.; Varambally, S.; Creighton, C.J. Pan-cancer molecular subtypes revealed by mass-spectrometry-based proteomic characterization of more than 500 human cancers. Nat Commun 2019, 10, 5679, doi:10.1038/s41467-019-13528-0.

  1. Chandrashekar, D.S.; Bashel, B.; Balasubramanya, S.A.H.; Creighton, C.J.; Ponce-Rodriguez, I.; Chakravarthi, B.; Varambally, S. UALCAN: A Portal for Facilitating Tumor Subgroup Gene Expression and Survival Analyses. Neoplasia 2017, 19, 649-658, doi:10.1016/j.neo.2017.05.002.

Additionally, the reference Agarwal et al., 2015 was added to the manuscript for citing the TargetScan tool, which is also incorporated in the UALCAN tool.

Agarwal, V.; Bell, G.W.; Nam, J.W.; Bartel, D.P. Predicting effective microRNA target sites in mammalian mRNAs. Elife 2015, 4, doi:10.7554/eLife.05005.

The conclusions are insufficient and too general. The conclusions should contain specific information. The author should summarize the results of his analysis and should indicate what the results of this analysis show (as evidenced by the research results, what the obtained knowledge contributes to the scientific field). 

Answer: conclusion were extended with summary of the databases and tools used for the analysis and main results.

Figure 3 is out of focus - please correct it.

Answer: Figure 3 was replaced with the new version with a better resolution.

Reviewer 2 Report

       The work in this manuscript is the first to bring together 18 different molecular interactions linked to the HIF1A gene. The HIF1A gene plays a role in the development of diseases like cancer, diabetes, and erythrocytosis.

 The overall findings of Tanja Kunej show that the current data synthesis on the HIF1A gene is the first step towards creating a map of regulatory factors for the HIF1A gene, with the intention of building a holistic perspective of the HIF1A gene by combining data from diverse genomics resources and publications.

The manuscript was organized well but there are few issues listed below that need to be addressed.

  • A graphical abstract at the end of the Discussion to sum up all the genomic databases and bioinformatics tools with a little information about each may further make the paper more reader-friendly.
  • The author may elaborate a little more on the Materials and Methods section describing all of this needs a little more details.
  • In line 160, the clarity of the presentation needs to be improved.
  • The author stated results in lines 241 -266. However, more details are needed so that the reader can understand the impact of this information.

Author Response

Reviewer 2

The work in this manuscript is the first to bring together 18 different molecular interactions linked to the HIF1A gene. The HIF1A gene plays a role in the development of diseases like cancer, diabetes, and erythrocytosis.

The overall findings of Tanja Kunej show that the current data synthesis on the HIF1A gene is the first step towards creating a map of regulatory factors for the HIF1A gene, with the intention of building a holistic perspective of the HIF1A gene by combining data from diverse genomics resources and publications.

The manuscript was organized well but there are few issues listed below that need to be addressed.

  • A graphical abstract at the end of the Discussion to sum up all the genomic databases and bioinformatics tools with a little information about each may further make the paper more reader-friendly.

Answer: The summary of genomic databases and bioinformatics tools used in the present analysis is presented in the Figure S2.

  • The author may elaborate a little more on the Materials and Methods section describing all of this needs a little more details.

Answer: The Materials and Methods section has been extended.

  • In line 160, the clarity of the presentation needs to be improved.
  • The author stated results in lines 241 -266. However, more details are needed so that the reader can understand the impact of this information.

Answer: The text has been extended and rewritten for clarity.

Round 2

Reviewer 1 Report

Please mark the individual parts of the figure S1 according to the legend (A, B, C, D).

It would be good to improve the sharpness of Figure 8.